# Remote inspection of adversary-controlled environments

Johannes Tobisch [1] ✉, Sébastien Philippe [2] ✉, Boaz Barak [3], Gal Kaplun[3], Christian Zenger [4,5], Alexander Glaser[2], Christof Paar [1] & Ulrich Rührmair [6,7] ✉

Remotely monitoring the location and enduring presence of valuable items in adversary-controlled environments presents significant challenges. In this article, we demonstrate a monitoring approach that leverages the gigahertz radio-wave scattering and absorption of a room and its contents, including a set of mirrors with random orientations placed inside, to remotely verify the absence of any disturbance over time. Our technique extends to large physical systems the application of physical unclonable functions for integrity protection. Its main applications are scenarios where parties are mutually distrustful and have privacy and security constraints. Examples range from the verification of nuclear arms-control treaties to the securing of currency, artwork, or data centers.

Remotely monitoring valuable items in adversary-controlled environments constitutes an intricate problem. Traditional inspection and surveillance methods are not always possible to implement or may fall short of meeting stringent security and privacy requirements. It may be difficult and perhaps impossible to permit regular physical inspections or placing CCTV cameras in such secure environments to offer some level of confidence in the integrity of stored items. Agreed managed-access inspections leave open the possibility of inspectors gathering information. Providing confidence that relevant surveillance data are originating from the correct location and have not been pre-recorded could prove challenging when the environment is controlled by an adversarial party[1,2]. In this context, specialized surveillance hardware and cryptographic tools are at risk of hacking and spoofing.

An example for this problem can be found in the monitoring of non-deployed strategic and tactical nuclear warheads as part of an arms-control agreement. These warheads represent 70% of the global nuclear arms stockpile and remain outside of existing agreements because of the difficulties to monitor them. They are stored in dedicated bunkers at sensitive military or nuclear sites[3]. The presence and number of such weapons at any given site cannot be verified easily via satellite imagery or other national technical means that are unable to see into the storage vaults. To include them in future arms control initiatives, there is a need to develop technologies and protocols to reliably assess if weapons declared as being in storage are not removed[4].

Here we propose and demonstrate a new remote monitoring approach based on a radio-wave measurement system to generate fingerprints of a room and its content using an array of randomly oriented mirrors to verify that nothing changes over time. This approach only requires a single on-site visit to initialize the monitoring protocol to install the mirrors and take an initial imprint of the room. Our approach builds on the concepts of physical unclonable functions[5,6] (PUFs) and virtual proofs of reality[7], for which data authenticity, confidentiality, and integrity does not rely on digital keys and algorithms but on the inherent material complexity of physical systems to achieve privacy and security objectives. Our work shows that large-scale systems such as an entire room and its content can also be turned into physical unclonable functions.

The basis of our monitoring scheme is fingerprint matching using a challenge-response protocol between two parties (a prover and a verifier) in two separate locations. Here, the prover owns a set of items

[1]Max Planck Institute for Security and Privacy, Bochum, Germany. [2]Program on Science and Global Security, Princeton University, Princeton, NJ, USA. [3]John A. Paulson School of Engineering and Applied Sciences, Harvard University, Boston, MA, USA. [4]PHYSEC GmbH, Bochum, Germany. [5]Secure Mobile Networking, Ruhr University Bochum, Bochum, Germany. [6]Electrical Engineering and Computer Science Department, TU Berlin, Berlin, Germany. [7]Secure Computation Laboratory, University of Connecticut, Storrs, Mansfield, CT, USA. ✉e-mail: johannes.tobisch@mpi-sp.org; sebastien@princeton.edu; ruehrmair@ilo.de

stored inside a room and must provide answers (or responses) to questions (challenges) asked by the verifier to demonstrate the undisturbed character of the storage space and its contents. The security of the protocol relies on the idea that the questions can only be answered correctly if the spatial and material configuration of the room remains unchanged.

## Results

### Verifying the integrity of valuable items in storage

To assess the integrity of items stored in a room, we rely on the emissions and measurements of radio signals between a transmitting and a receiving antenna (Fig. 1). The complex multipath propagation of the signal within the room provides a unique and reproducible fingerprint of its spatial configuration. We complement this measurement system with a challenge generation mechanism comprising a set of radio-wave-reflecting mirrors that can be individually and reproducibly rotated. Each channel response measurement between a pair of antennas becomes a convolution of the pilot signal, the combination of individual mirror rotations (or challenge), and the static components of the room. Any modification, such as the displacement or the removal of an item, alters the multipath propagation of the pilot signal resulting in a different measured response.

The inspection protocol consists of an on-site setup phase and a remote proof phase (Fig. 1b). During the setup phase, the verifier installs the challenge-response system in the room, which allows the collection of challenge-response pairs both during the setup and during the proof phase. Once the challenge-response system is installed, the room is sealed. To complete the setup phase, the verifier randomly chooses a sufficiently large set of challenges and collects the

corresponding responses. The prover must not learn this list of secret challenges.

Once the verifier has collected the challenge-response pairs, the protocol enters the long-term proof phase. From now on, the verifier is remote and regularly sends challenges drawn from her secret list to the prover. He must then send back the corresponding responses within an agreed short time interval. If a response is close to the original, the verifier accepts the proof and deletes the challenge-response pair from her list to prevent re-use. The time between two queries has to be short enough, perhaps on the order of a minute, to avert tampering efforts by the prover between queries. These efforts can be further hampered by the introduction of fake queries, i.e., the verifier continuously sends random challenges interspersed with those from the secret list. Only the responses to the latter challenges can actually be verified. The prover, however, cannot distinguish whether a particular challenge belongs to the secret list because all challenges are drawn uniformly random.

From a security point of view, the prover should (1) not be able to alter the state of the room in a manner that cannot be detected, (2) not be able to collect all the responses for the entire challenge space, and (3) not be capable of predicting responses for arbitrary challenges by either building a functionally identical room or producing a mathematical model of the function mapping challenges to responses. We address these security requirements one-by-one in the following, leveraging experimental data and results.

### Experimental realization

To provide a proof of principle, we built a prototype challenge-response system installed in a metal storage container

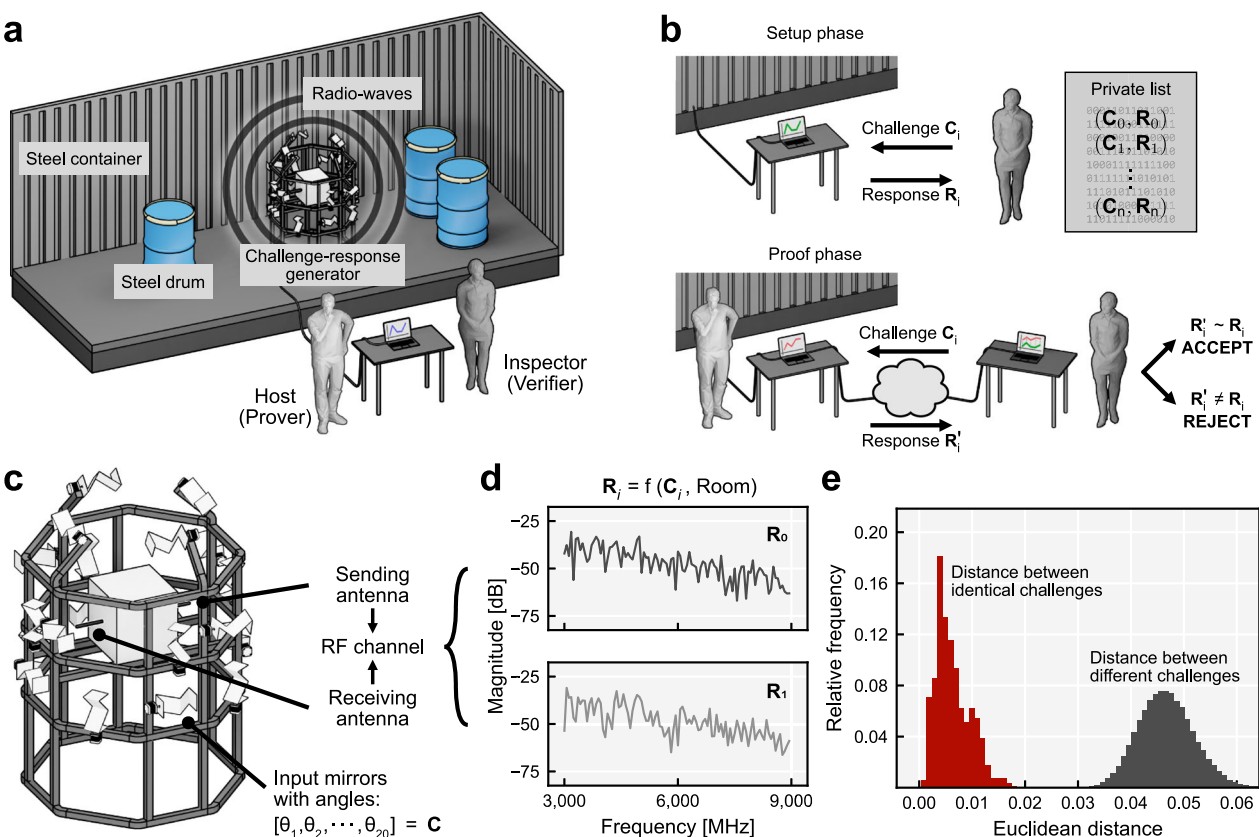

**Fig. 1 | Inspection protocol and experimental realization. a** Complete experimental setup housed in a steel container. **b** The protocol consists of an on-site setup and a remote proof phase. **c** A challenge C consists of the rotational angle of all 20 mirrors. **d** Radio-frequency responses for two random challenges. **e** The measurement error (intra distance, red) over a 4-week measurement campaign in comparison to the inter-challenge response diversity (red).

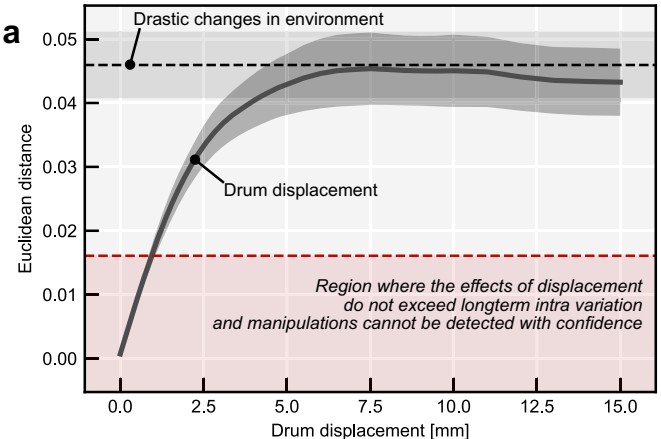
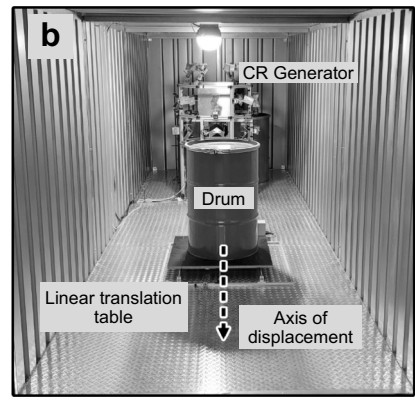

**Fig. 2 | Detecting minute physical changes in the room. a** Solid gray curve shows the mean effect of drum displacement on responses. Black dashed line shows the mean effect of removing a drum from the container. Mean and standard deviation (shaded area) were obtained from measurements over 100 challenges. Red dashed line shows the mean plus three times the standard deviation of the long-term intra distance. **b** Overview of experimental setup, translation table in the front, and challenge-response (CR) apparatus in the back.

(6m × 2.2m × 2.1m) housing a number of empty 55-gallon steel drums. The challenge device consists of twenty small aluminum mirrors mounted in different orientations on separate stepper motors (see Fig. 1c and "Methods"). The mirrors can be rotated around one axis by 360° in 1.8° steps, yielding 200 possible positions per mirror. The challenge **C** that can be applied to the system encodes the position of the mirrors as a vector of 20 elements, where each element takes an integer value between 0 and 199. This results in notionally large challenge space of $200^{20} \approx 10^{46}$ different mirror configurations. For the response measurement, we use a vector network analyzer to measure the amplitude of the forward transmission parameter between two antennas. This parameter describes the attenuation of a signal at a specific frequency.

For a single response vector **R**, we collect the forward transmission parameter for 100 points that are evenly spaced between 3 and 9 GHz, which corresponds to a wavelength between $\lambda_{\min} \sim 3.3$cm and $\lambda_{\max} \sim 10$cm. Figure 1d shows sample responses for two different challenges. Together, a challenge and a response form a challenge-response pair (CRP).

The channel response is affected by measurement noise and shifts in environmental variables such as temperature and humidity, which means that a challenge does not map to a single discrete response, but to a continuum of possible responses. It is important that the intra distance, i.e., the variations between responses for the same challenge, remains low enough so that responses still act as fingerprints and uniquely reflect the applied challenge and the configuration of the room. Additionally, the inter distance between responses stemming from different, uniformly drawn challenges must be much larger to guarantee discriminability. To measure the distance between two response vectors **a** and **b**, we use the Euclidean distance defined as $d(\mathbf{a}, \mathbf{b}) = \sqrt{\sum_i (a_i - b_i)^2}$. Figure 1e shows the clear difference between the intra distance distribution and inter distance distribution based on measurements that were taken over a period of 4 weeks (see Supplementary Figs. 1–3 for drift over time and environmental variations in temperature and humidity). This shows that it is highly unlikely that any two randomly-drawn challenges will result in similar responses.

## Security of the system

Our inspection system must be sensitive to adversarial physical manipulations of the room and its content. In principle, our apparatus allows us to capture changes on the order of the pilot signal wavelength $\lambda$. To verify this property, we placed one of the drums on a linear

translation table (see Fig. 2b and "Methods"). Figure 2a shows that a displacement of ∼1mm leads to a response mismatch greater than three times the standard deviation from the mean of the long-term intra distance distribution. Furthermore, after a small displacement of around 7.5mm, corresponding to ∼$\lambda_{\mathrm{mean}}/10$, a first local maximum of the decorrelation is reached which is equivalent to the removal of a drum from the container.

For our system to be secure, the effective challenge space must be large enough to prevent the prover from building an exhaustive list of all existing challenge-response pairs. Such a brute-force attack would allow the prover to present a response from a database instead of a live measurement during an inspection query. In the case of our experimental setup, the rotation of a single mirror by 44° leads to a response change that is long-term detectable by the verifier ($3\sigma$ from the intra distance mean as shown in Supplementary Fig. 4). Hence, each mirror has at least $360/44 \approx 8$ effective challenge positions. The number of challenges increases exponentially with the number of mirrors. For our proof of principle comprising 20 mirrors, the total number of challenges is estimated to be greater than $8^{20} (\sim 10^{18})$, an amount that cannot be exhaustively queried.

Another important requirement for the security of the protocol is that the room and its content cannot be cloned. This means that it must be hard to create a physical replica that could be used to generate valid responses, even after the integrity of the original room has been violated by a malicious prover. Creating an exact copy of any environment that produces the same radio-frequency fingerprint as the reference is an inherently difficult task that can be made even harder by introducing unique objects into the room. Such complex objects could be brought by the inspector to the host facility and could even be produced in their final form shortly before the setup phase begins and the room is sealed. The latter could be realized, for instance, by crumpling aluminum foil into spheres or by bending stochastic metallic foam panels. For some applications, it could also be possible to set up a "room within a room"[8], for example by using a dedicated storage container with unique features prepared in a staging area and installed in a location of the prover's choice.

As opposed to a physical copy, an attacker could try to generate a digital clone of the room to predict responses during the proof phase. Here, we are concerned with two types of attacks. First, the possibility of running a full-scale electromagnetic (EM) simulation of the room within the time allotted to provide a response to a given challenge. Second, the possibility of breaking the protocol using an accurate machine-learning-based representation of the function mapping challenges to responses.

In the first case, the prover would need to develop a 3D model of the room capturing both its spatial configuration and material properties (i.e., permittivity, permeability, and conductivity). EM simulations of our "room" would require meshes with size $\sim\lambda/100$ to guarantee stability and accuracy of the solutions (including near field effects close to the antennas)[9]. Even for the relatively small-sized room used for our experiment ($60\lambda_{max} \times 22\lambda_{max} \times 20\lambda_{max}$), the attacker would need to solve, within a few seconds, a problem involving $\sim 1$ trillion of unknowns, which is three orders of magnitude larger than current state-of-the-art large-scale EM simulations run on supercomputers[10,11]. While mesh optimization and domain decomposition techniques could be used to speed up calculations, they would be inherently limited by the complexity of the problem. For example, the boundary conditions could be made arbitrarily complex by introducing aperiodic metallic structures (e.g., stochastic foams[12]) in the room, making this attack vector impractical.

Ray tracing techniques[13] pose an alternative to full EM simulations. In our environment, we have determined a coherence bandwidth of $\sim 1$ MHz (see Supplementary Figs. 7 and 8), which is indicative of a high reflectivity and the dominance of small-scale fading effects. A major use case of ray tracing is the approximate prediction of path loss in large outdoor[14] and indoor[15] environments with simplified geometry, which does not readily transfer to our highly reflective environment and its varied frequency response.

In the second case, the attacker's goal is to derive a mathematical model that accurately maps challenges to responses on the basis of a given training set of challenge-response pairs. The attacker model is the same as in the case of Strong PUFs[16–19] while the concrete learning problem that needs to be solved differs in our scenario. Here we consider that the system is compromised if the mean prediction error falls under the threshold of the intra distance mean plus $3\sigma$. This conservative definition favors the attacker. In practice, the distribution of the prediction error would need to be much closer to the intra distance to consistently fool the verifier into accepting forged responses for the duration of the proof phase.

To explore the viability of this attack and how the prediction error scales with the number of input mirrors, we trained multiple machine learning models, including linear regression, *k*-nearest neighbors, gradient-boosted trees, and neural networks (see Supplementary Fig. 5). We used training sets of up to 1,280,000 challenge-response pairs each corresponding to a number of active mirrors ranging from 4 to 20, requiring continuous measurements for up to seven days. We obtained the best predictions with a densely connected neural net (consisting of 8 hidden layers with 3072 neurons each) beating among others gradient boosted trees[20,21]. Our results show that the neural net ability to defeat the system is both a function of the numbers of mirrors and the training set size. For a number of mirrors in the 16–20 range and a $10^6$-CRPs training set, our neural net could not fool the verifier in accepting synthetic responses.

To better understand how security scales up, we computed learning curves that show the dependency between the training set size and the learning error (Fig. 3). From our experimental data, we find that the learning error follows a power-law type dependency. Using this scaling law, we find that breaking systems of 16 and 20 active mirrors would require training set sizes of $\sim 3,000,000$ and $\sim 11,000,000$ acquired over $\sim 2$ weeks and $\sim 8$ weeks of continuous measurements, respectively. Similar power laws can be observed in other deep learning domains and it is assumed that advances in neural net design only improve the scaling by a constant factor[22,23].

Our experimental data also suggests that the required training set size scales polynomially with the number of active mirrors following a power law (Supplementary Fig. 6). The attacker is fundamentally limited by the fact that the training set acquisition cannot be parallelized and the maximum amount of data that can be collected is determined by the per-response acquisition speed and the total duration of the

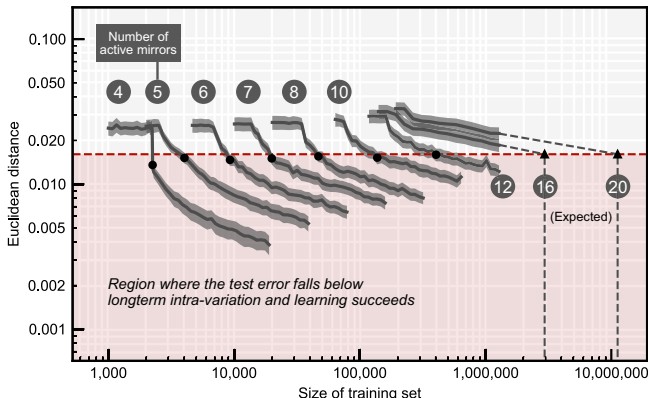

**Fig. 3 | Robustness against machine learning attacks.** Solid curves show the mean prediction error for 1000 test samples for our neural network architecture over the number of training samples. Dots and triangles mark training set size at which the system is broken. Both axes are log-scaled. Gray dashed lines are log-log extrapolated. Gray shaded areas mark standard deviation. Red dashed line shows the mean plus three times the standard deviation of the long-term intra distance.

protocol. The number of mirrors thus acts analogously to a security parameter in classical cryptographic applications that needs to be tuned to provide an acceptable security margin, taking potentially improved machine learning models into account. We would like to stress that the number of mirrors is not limited to 20.

## Discussion

We have experimentally demonstrated key aspects of a radio-wave-based inspection system designed to remotely confirm that valuable items located in an adversary-controlled environment are kept in storage. Our proof of principle does neither require secure communication channels nor tamper-resistant sensor hardware at the inspected site. Our inspection system yields reproducible results in an extensive measurement campaign for a notional storage facility, can detect physical changes on the order of a few millimeters, and is robust against major physical and computational attacks. In addition, important security parameters such as the number of input mirrors, the wavelength of the probing signal, and the complexity of the room can be scaled to the disadvantage of potential attackers. The baseline scenario for which we intend this concept addresses long-established concerns with the verification of non-deployed nuclear weapons, where parties are mutually distrustful, have privacy and security constraints, and want to keep the interval of on-site visits at sensitive facilities to a bare minimum. Beyond nuclear arms control verification, our inspection system could find application in the financial, information technology, energy, and art sectors. The ability to remotely and securely monitor activities and assets is likely to become more important in a world that is increasingly networked and where physical travel and on-site access may be unnecessary or even discouraged.

## Methods
### Experimental setup
The complete experimental system is housed in a flat pack steel container that has a length of 6 m, a width of 2.2 m and a height of 2.1 m, located indoors at the Bochum Max Planck Institute. The challenge-response system is mounted on an aluminum strut frame. The frame has an octagonal base with a side length of 36 cm and a height of 150 cm. An aluminum-plated hollow cube with an edge length of 40 cm is attached at the center and houses electronic control and data acquisition equipment. Two Taoglass FXUWB10.07.0100 C wideband antennas are mounted on opposing sides of the cube such that no direct line of sight exists between them. Twenty mirror assemblies are

attached to the frame and surround the cube. Each mirror assembly comprises a NEMA 17 stepper motor, an infrared end switch to identify the zero position, and a $\Sigma$-shaped aluminum mirror that has a height of 16.5 cm, a width and length of 8 cm and a plate thickness of 1 mm. The motors are controlled via STMicro L6470 chips and allow for 360° rotation in 1.8° incremental steps.

The two antennas are connected to a Keysight P9372B Vector Network Analyzer (VNA).

With the VNA, we measure the magnitude of the complex forward transmission coefficient (scattering parameter $S_{21}$) for 100 equally spaced points in the frequency range from 3 to 9 GHz, using an IF bandwidth of 30 kHz, unless otherwise stated. Setting a challenge and collecting its corresponding response takes in total $\sim 0.4$ s.

### Acquisition of challenge-response pairs
In all our experiments, we sample challenges using a uniformly random distribution (unless stated otherwise). For the drum displacement and removal experiments (Fig. 2b), we used a set of 100 individual challenges. We started by setting the linear translation table to its zero position and acquiring a reference measurement for each challenge. Then we moved the drum in 0.25-mm increments using the linear translation table. We doubled the step size to 0.5 mm after reaching a displacement of 5 mm. After 10 mm, we moved the drum in millimeter steps until reaching a total displacement of 15 mm. At each step, we measured new responses for each challenge. Once the final position was reached, the linear table returned to its zero position. Finally, we removed the drum from the container and collected one last set of responses. To study the effect of mirror rotation (Fig. 2d), we selected 100 challenges and a single active mirror. For each challenge, the 19 inactive mirrors remained in position, while the active mirror was rotated in 1.8° steps up to 90°. This process was repeated for each of the 20 mirrors.

To train our machine learning models (Fig. 3), we collected challenge-response pairs for increasing numbers of active mirrors (4, 5, 6, 7, 8, 10, 12, 16, and 20). In each case, we collected a test set and a training set, whose sizes range from 20,000 to 1,280,000. In total, we collected more than 5,000,000 challenge response pairs. All inactive mirrors were kept at a static default position during the duration of the experiment.

Additionally, a smaller set comprising of 25 challenges was measured in an interleaved fashion with the training set and test set (measured once every 1000 challenges of the training set). The room was not disturbed in between experiments and we use this data from all ML sets to compute the intra distances.

### Determination of coherence bandwidth
For the results shown in Supplementary Figs. 7 and 8, we have collected responses for 100 random challenges at 60,000 equally spaced points in the frequency range from 3 to 9 GHz, i.e., sampled points lie 100 kHz apart from each other. Here, we collected the full complex scattering parameter $S_{21}$ instead of only the amplitude.

We have computed the autocorrelation for the set of 100 responses, i.e., the correlation between the response and a frequency-shifted version of itself [24].

### Machine learning models
For our proof of principle, a machine learning attack is a multi-output regression problem, i.e., the prover uses a learning algorithm to produce a function that maps challenges (up to 20 mirror rotations) to the real amplitudes of the response vector (each response vector having $100$ elements). In this work, we evaluated four types of algorithms including linear regression, $k$-nearest neighbors, neural networks, and gradient-boosted trees.

Linear regression serves as a baseline to rule out the applicability of a simple linear model. The k-nearest neighbors algorithm works

conceptually similar to building a look-up table of examples and its ineffectiveness supports our analysis of the effective challenge space size. With gradient-boosted trees and neural networks, we have explored two techniques that are very successfully used on many diverse real world data sets. The evaluation included a hyper-parameter exploration for each algorithm. Among the four candidates, neural networks performed the best.

For each model, we used the mirror rotation angles in Cartesian coordinates as the function input. For linear regression, we used the Linear regression implementation from Scikit-learn [25] (Version 0.23.1), with default parameters. For $k$-nearest neighbors, we used the KNeighborsRegressor implementation from Scikit-learn (Version 0.23.1). We used default parameters and varied the number of neighbors parameter in the range from 2 to 250. The best result was achieved for 31 neighbors.

The neural network was implemented in PyTorch [26]. For the architecture, we chose a series of densely connected layers, rectified linear activation function (ReLU), and layer normalization. We set the number of hidden layers to 8 and the number of neurons per layer to 3072. The networks were trained with the Adam optimizer and a learning rate of 0.0002, for 400 epochs and with a batch size between 64 and 4096. For gradient-boosted trees, we used the XGB implementation (Version 1.1.1). We used early stopping to determine the optimal number of estimators and exhaustively tested for the optimal tree depth between 2 and 8.

### Data availability
The raw data (challenge and responses) have been deposited in the open research data repository of the Max Planck Society at https://doi.org/10.17617/3.UNYGGC. Source data are provided with this paper.

### Code availability
All code supporting the findings of this article are available from the corresponding authors upon request.

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

## Acknowledgements

A.G. and S.P. acknowledge funding from U.S. DOE/NNSA (DE-NA 0002534) and the Carnegie Corporation of New York. C.P. and J.T. acknowledge support from the DFG Excellence Strategy grant 39078197 (EXC 2092, CASA). B.B. and G.K. were supported by a Simons Investigator Fellowship, NSF grant DMS-2134157, DARPA grant W911NF2010021, and DOE grant DE-SC0022199. U.R. acknowledges funding by the AFOSR award FA9550-21-1-0039.

## Author contributions

J.T. and S.P. wrote the first draft of the manuscript and share first authorship. J.T. and S.P. conceived, planned, and analyzed results from the experiments, with ideas and feedback from C.Z., C.P., A.G., and U.R. J.T. carried out the experiments. J.T., G.K., and B.B. trained machine learning algorithms. All authors contributed to the development of the inspection protocol. U.R. first proposed the concept of Virtual Proofs of Reality in facility protection, on which this research is based. U.R. and C.Z. proposed the idea of combining physical-layer security and Virtual Proofs of Reality. C.Z. and C.P. developed the first prototype of the inspection system, with inputs from U.R. U.R. and S.P. initiated the project, and C.P. and A.G. secured funding for the experimental work. All authors provided critical feedback and helped shape the research, analysis, and writing of the manuscript.

## Funding

## Competing interests

The authors declare no competing interests.
