## [Peer Review File · Nature Communications]

Remote Inspection of Adversary-Controlled EnvironmentsREVIEWER COMMENTS

Reviewer #1 (Remarks to the Author):

This paper designed a novel algorithm to verify whether the environment and the properties inside the environment have been changed adversarially. The authors borrowed the concept of challenge-response-pair used in PUF and leveraged the unique characteristics of wireless propagations. The idea is novel and I believe it is the first time to do so.

What are the challenge-response pairs in this paper? The authors did mention “each response vector having 100 elements” in line 253, but what is it exactly? Can the authors give some examples of the challenge-response pairs? The reason I am asking is that wireless communication systems are different from digital circuit-based PUF, where both challenges and responses are digital bits. In wireless systems, the payload (digital bits) is modulated and transmitted over wireless channels (in analog form). The signals are captured by a receiver (in analog form) and demodulated to digital bits. What data/information is leveraged by the authors? What wireless standards/modulations are used? Will the adopted wireless technique affect the algorithm in this paper?

The theoretical aspect of the paper is limited. In particular, it is essential to ensure there is uniqueness among challenge-response pairs. In other words, one challenge should map to a unique response. While there are some empirical results provided, there is no theoretical analysis or discussion.

Regarding the analysis provided in line 145-line 157, there are ray tracing models and software available. Can the authors confirm if they have looked into any of the existing software and give some results that the system is secure against commercial ray tracing software?

Regarding the machine learning part, from line 165 to line 174, is the description about the prover? Why the prover is considered as an attacker? I recommend providing a clear explanation of what are the prover and verifier. In addition, the machine learning algorithms adopted by the authors seem quite simple to me. Have the authors considered using

sophisticated deep learning techniques? Finally, I believe a reference regarding machine/deep learning attacks against PUF will be helpful.

In line 31, the authors mentioned: "As an example of this approach, we consider its application for the monitoring of non-deployed strategic and tactical nuclear warheads as part of an agreement." However, it seems the results are not from this application.

Reviewer #2 (Remarks to the Author):

The concept of this work is quite interesting. It combines the PUF and virtual proof of reality to verify the location and enduring presence in a container (e.g., room). This later be viewed as a new hardware security primitive without relying on any conventional 'secrecy' in cryptography.

The proof of concept is performed through using the radio-wave scattering pattern represented response conditioned the querying challenge. For application scenarios, it is motivated from the nuclear weapon control.

The presentation is clear, the reviewer enjoys the reading. The protocol is firstly provided. Then the authors show how to achieve the target step-by-step (e.g., increasing the CPR space to be exhaustively characterized in reasonable time, showing the intra-variance is much smaller than the inter-variance, the challenge of mathematically modeling the PUF and the EM simulation difficulty that is originated from SHIC PUF concept). All of these together make an interesting and reasonable study, where each step is reasonably explained/validated.

The writing is good and easy to follow.

The reviewer has reviewed a number of related works on PUF submitted to NC and feels this is the best one so far for its interesting conceptualization, motivation and experimental validation, as well as the easy presentation style.

One of 'onsite' and 'on-site' can be chosen for consistency usage, similar for others e.g., setup vs set-up.

Point by point responses to reviewers' remarks

July 14, 2023

Reviewer #1

This paper designed a novel algorithm to verify whether the environment and the properties inside the environment have been changed adversarially. The authors borrowed the concept of challenge-response-pair used in PUF and leveraged the unique characteristics of wireless propagations. The idea is novel and I believe it is the first time to do so.

Response: We thank the reviewer for recognizing the novelty of our work and providing thoughtful remarks that we address below and in key revisions of the paper. We believe that the manuscript has improved as a result.

What are the challenge-response pairs in this paper? The authors did mention “each response vector having 100 elements” in line 253, but what is it exactly? Can the authors give some examples of the challenge-response pairs?

Response: A challenge consists of a set of rotational positions (or simply “angles”) of the 20 mirrors in our setup (see Figure 1a and 1c). Mathematically speaking, a challenge can be viewed as a 20-tuple $(\theta_1, \dots, \theta_{20})$. Each mirror can be rotated in steps of 1.8° by our specific hardware, leading to $360^\circ/1.8^\circ = 200$ possible values for each θ_i . For the entirety of 20 independent mirrors, this results in $200^{20} \sim 10^{46}$ nominally possible challenges.

Our room presents a rich multi-path environment, which means that a radio signal transmitted between two antennas will be subject to attenuation and a phase offset. For our response we measure the attenuation. More specifically, we measure the absolute value of the complex forward transmission coefficient (scattering parameter S_{21}) for 100 frequencies that are evenly spaced between 3 and 9 GHz. Taken together, these values make up our response vector of 100 elements.

Specific changes: We clarified these aspects in the main text. A full description is given lines 243–246 in the Methods section. We provide examples of responses for given challenges in Figure 1d. We have added a more detailed description of the challenges and responses in lines 85–97 and referenced Figure 1c in the main text.

The reason I am asking is that wireless communication systems are different from digital circuit-based PUF, where both challenges and responses are digital bits. In wireless systems, the payload (digital bits) is modulated and transmitted over wireless

channels (in analog form). The signals are captured by a receiver (in analog form) and demodulated to digital bits. What data/information is leveraged by the authors? What wireless standards/modulations are used? Will the adopted wireless technique affect the algorithm in this paper?

Response: In our system, no standard digital data nor any other message in the classical sense is transmitted between the sending and receiving antennas. Hence, our setup does not form a wireless communication system.

Instead, it works as follows: The challenge configures the positions of the mirror ensemble in the room. Subsequently, our sending antenna always emits the same, fixed waveform (a sinusoid), which is independent of the applied challenge. It is individually shaped only when hitting and traversing through the challenge-configured mirror ensemble. The transfer function of the entire room is consequently determined by (i) the flexible, challenge-configured mirror ensemble within the room, and (ii) the fixed, static “rest” of the room.

The receiving antennas finally measure the response of the room. This measurement is done using a vector network analyzer. It essentially sweeps the selected frequencies and observes the frequency shift and attenuation of a sinusoidal signal for each frequency point. Among other things, this makes our approach relatively easy to apply in practice.

The theoretical aspect of the paper is limited. In particular, it is essential to ensure there is uniqueness among challenge-response pairs. In other words, one challenge should map to a unique response. While there are some empirical results provided, there is no theoretical analysis or discussion.

Response: We agree with the reviewer that it is fundamental to ensure uniqueness among challenge-response pairs. This is why we designed a comprehensive measurement campaign to provide a lower bound number of independent pairs.

Our analysis is detailed (i) in the presentation of the so-called “inter-distance” in Figure 1e, and (ii) in our discussion of the challenge space size in lines 122–132. The experimental data demonstrates that it is extremely unlikely that two random challenges would produce the same responses.

Specific changes: We clarified this last point in the main text (line 110–111).

Our analysis follows the usual methodology in the Physical Unclonable Functions literature. For example, our approach is very similar to the analysis of the “inter-chip” variation, i.e., the uniqueness within a PUF population (see for example Roel Maes, Physically Unclonable Functions: Constructions, Properties and Applications).

This experimental validation is fundamental for any real-world uses of the scheme.

Regarding the analysis provided in line 145-line 157, there are ray tracing models and software available. Can the authors confirm if they have looked into any of the existing software and give some results that the system is secure against commercial ray tracing software?

Response: First, as we discuss in our security analysis in lines 149–172, a precise simulation of the entire room for adversaries, but also for any other parties (including the authors), is practically impossible.

It is not feasible for us to conduct exhaustive and exact room simulations. The difficulty of creating a digital clone is an important security feature of our scheme.

To conduct any simulations, an attacker would need to accurately represent the boundary conditions of the room and its content. As we discuss in the paper, it is possible for the inspector to introduce complex metallic objects, such as stochastic metallic foams or other unique aperiodic structures, in the room that would make the boundary conditions of any simulations arbitrarily complex without the possibility of full characterization.

While ray-tracing techniques would simplify numerical simulations, they would do so at the cost of reduced accuracy. This would likely introduce systematic errors in any responses.

Specific changes: On the basis of new data, we demonstrate the high reflectivity of and small-scale fading effects within the room by estimating the coherence bandwidth which lies in the range of ~ 1 MHz (See new Supplementary Figures 7 and 8).

Taken together, this result and the ability of the inspector to complexify boundary conditions arbitrarily underline the difficulty of any simulation approaches, including ray-tracing methods that are usually used to estimate coarse-grained channel properties in simplified scenarios.

Regarding the machine learning part, from line 165 to line 174, is the description about the prover? Why the prover is considered as an attacker? I recommend providing a clear explanation of what are the prover and verifier.

Response: Our use of “prover” and “verifier” follows the terminology established in the cryptography and security communities. The two terms are routinely used for (both PUF-based and classical) identification protocols and also in the context of Virtual Proofs of Reality on which this paper is based (See ref 5 in the paper).

In our case, the prover is an entity that wants to convince another entity, the verifier, remotely of the claim that a certain room or container at the prover’s location has not

been changed or altered. In our protocol, it is assumed that the verifier has had the chance to inspect the room/container once before the protocol starts.

A key aspect of our systems is that we cannot assume that the prover is honest, though: The prover has an incentive to cheat, i.e., the room has been tampered with and the prover wants to trick the verifier into falsely accepting that the room is unaltered. From a security standpoint, we thus need to consider the case of the prover being an adversary. As a result, we consider relevant attacks (see lines 74–80).

In addition, the machine learning algorithms adopted by the authors seem quite simple to me. Have the authors considered using sophisticated deep learning techniques?

Response: For our security analysis, we applied well-known machine learning techniques including a deep neural network, which was the most performant algorithm.

The results presented in Figure 3 illustrate our effort at modeling the system. To obtain these learning curves, we had to train our neural network (comprising 8 layers with 3072 neurons per layer, more than 75 million parameters in total) 12 to 40 time on up to 1.2 million challenge-response pairs acquired experimentally to obtain each individual curve (taking weeks of GPU time). To make this training possible, we had to collect ~5 million challenge response pairs, which took ~30 days of continuous experiments.

Our point here was to characterize the hardness of the learning problem as a function of the number of mirrors.

Within the class of neural networks and the context of deep learning, there are a lot of domain-specific architectures, such as convolutional nets in image classification or transformers/attention mechanisms in large language models. These domains are quite different from the learning problem at hand: data is of high dimension and carries internal structure such as being sequential or being invariant to rotation, translations and scaling. In our case, the low-dimensional inputs (the challenges) to the model are i.i.d. drawn values and the order of mirrors within the challenge vector is arbitrary. Existing advanced domain-specific architectures that target high-dimensional data with internal structure cannot be directly applied here.

Furthermore, the current understanding within the machine learning community is that while architectural modifications can enhance performance on a fixed dataset size, the loss curves exhibit power-law behavior with consistent exponents when comparing models across varying dataset sizes (See for example, in Figure 7 of the classical paper by Kaplan et al. from 2020 cited in reference 22 of the paper). Consequently, even if an attacker identifies a yet further optimized architecture for a certain task, the improvement would only be a constant factor. We now make this point explicitly in our analysis.

Our existing ML-analysis indeed shows that the scaling of the required number of ML-training examples is polynomial in the number of mirrors. Therefore, this constant factor can only provide a limited advantage for a fixed mirror count. This makes the number of mirrors similar to a classical “security parameter” that can easily be further increased, should the necessity ever arise in future applications. Increasing the number of mirrors from 20 (as in our submission) to 30 would have a drastic effect on the required training set size (or ML complexity). Given that the attacker can only gather a fixed amount of training data during the protocol runtime, a sufficient security margin can be achieved.

Specific changes: We have added a sentence regarding the influence of neural network architecture on the scaling (lines 196 to 198), changed the wording regarding the number of mirrors as a security parameter (lines 204 to 207) and described the reasoning behind our choice of algorithms (lines 284 to 289).

Finally, I believe a reference regarding machine/deep learning attacks against PUF will be helpful.

Response: Thank you for this suggestion. We have added an additional reference (Santikellur et al. Deep Learning based Model Building Attacks on Arbiter PUF Compositions). In their work, neural nets that are similar in design but smaller than our models have been used to learn Arbiter PUF-based constructions. We would like to note that the bulk of the literature describes attacks on such Arbiter PUF-based designs. Here, the learning problem is different from our case (classification task instead of regression). In particular, a concise mathematical model of these PUFs is known and forms the basis of all effective attacks (even in the case of deep learning, which requires a model-derived feature engineering step). As a consequence, specific attack techniques tailored for these silicon Strong PUFs cannot be applied in our scenario.

In line 31, the authors mentioned: “As an example of this approach, we consider its application for the monitoring of non-deployed strategic and tactical nuclear warheads as part of an agreement.” However, it seems the results are not from this application.

Response: Our approach was first designed with this application in mind. It is of course difficult to run our proof-of-concept experiments in a real military storage facility with live nuclear warheads. Our approach thus follows and mimics this situation. We demonstrate that our method would work in, and carry over to, such a setting, and lead a full proof of concept. It is our experience that national laboratories that have the possibility to run experiments in real conditions tend to do so when new promising methods are developed and we hope that it will be the case with this work.

This, of course, holds for any other application mentioned in the manuscript (protecting data centers, storage vaults, art works, currency, etc.). In the end, our goal was to provide a robust general demonstration of our technique.

We thank the reviewer again for its careful reading of the manuscript and comments, which have helped us improve our results and clarify our analysis.

Reviewer #2

The concept of this work is quite interesting. It combines the PUF and virtual proof of reality to verify the location and enduring presence in a container (e.g., room). This later be viewed as a new hardware security primitive without relying on any conventional 'secrecy' in cryptography.

The proof of concept is performed through using the radio-wave scattering pattern represented response conditioned the querying challenge. For application scenarios, it is motivated from the nuclear weapon control.

The presentation is clear, the reviewer enjoys the reading. The protocol is firstly provided. Then the authors show how to achieve the target step-by-step (e.g., increasing the CPR space to be exhaustively characterized in reasonable time, showing the intra-variance is much smaller than the inter-variance, the challenge of mathematically modeling the PUF and the EM simulation difficulty that is originated from SHIC PUF concept). All of these together make an interesting and reasonable study, where each step is reasonably explained/validated.

The writing is good and easy to follow.

The reviewer has reviewed a number of related works on PUF submitted to NC and feels this is the best one so far for its interesting conceptualization, motivation and experimental validation, as well as the easy presentation style.

Response: We are honored by the reviewer's very positive feedback. Thank you!

One of 'onsite' and 'on-site' can be chosen for consistency usage, similar for others e.g., setup vs set-up.

Response: Thank you for catching this. We made the use of the term consistent throughout the entire manuscript.

REVIEWERS' COMMENTS

Reviewer #1 (Remarks to the Author):

The authors have addressed my concerns. I have no further comments.